# A Gaussian to Vector Vortex Beam Generator with a Programmable State of Polarization

**DOI:** 10.3390/ma15217794

**Published:** 2022-11-04

**Authors:** Jacek Piłka, Michał Kwaśny, Adam Filipkowski, Ryszard Buczyński, Mirosław A. Karpierz, Urszula A. Laudyn

**Affiliations:** 1Faculty of Physics, Warsaw University of Technology, Koszykowa 75, 00-662 Warsaw, Poland; 2Photonic Materials Group, Łukasiewicz Research Network-Institute of Microelectronics and Photonics, Al. Lotnikow 32/46, 02-668 Warsaw, Poland; 3Faculty of Physics, University of Warsaw, Pasteura 5, 02-093 Warsaw, Poland

**Keywords:** optical vortices, structured light polarization, beam polarization conversion

## Abstract

We study an optical device designed for converting the polarized Gaussian beam into an optical vortex of tunable polarization. The proposed device comprised a set of three specially prepared nematic liquid crystal cells and a nano-spherical phase plate fabricated from two types of glass nanotubes. This device generates a high-quality optical vortex possessing one of the multiple polarization states from the uniformly polarized input Gaussian beam. Its small size, simplicity of operation, and electrical steering can be easily integrated into the laboratory and industrial systems, making it a promising alternative to passive vortex retarders and spatial light modulators.

## 1. Introduction

Optical vortices are states of light characterized by a phase rotation alongside the propagation axis resulting in the appearance of phase singularity and doughnut-like intensity profile [1]. The vortex beams themselves are known to carry an orbital angular momentum due to the azimuthal phase gradient and the subsequent tangential component of the Poynting vector [2,3,4]. The most fundamental representation of such beams is a scalar optical vortex, i.e., a beam with uniform polarization and phase continuously varying from zero to 2π*m* along its azimuthal coordinate. The number *m* denotes the helicity (or vorticity) parameter, known more commonly as the vortex’s topological charge (TC). For typical polarization states, such as linear or circular, the direction of the electric vector is uniform within the cross-section of a beam. In recent years the concept of complex optical beams has been extended to include waves with an inhomogeneous polarization structure, the so-called vector vortex beams, which are optical beams with spatially variant polarization as the azimuthal or radial, simultaneously maintaining a singularity in a phase and polarization [4,5]. For vector beams, the state of polarization (SOP) is space-variant. While radially and azimuthally polarized vector beams have an identical intensity distribution, their space-variant polarizations are orthogonal. Therefore, these beams behave quite differently in the photonic media and optical devices whose operation principle is polarization sensitive. One of the most striking examples of such polarization dependence involves the tight focus of vector beams. The radially polarized beam can be focused to a tighter focal spot than a linearly polarized beam of the same intensity distribution and acquires a transverse size below the diffraction limit [6,7].

Moreover, when tightly focused, this type of beam also experiences a transformation of its polarization state. It develops a strong longitudinal component of the electric field, transforming itself into the so-called needle beam. They are characterized by a focal pattern with tight beam size and extended depth of focus [8,9], which is of great significance in many applications, such as optical data storage, photolithography, and super-resolution microscopy. On the other hand, while having the same intensity distribution as the radially polarized beam, the one with azimuthal polarization retains its transverse character, even in a tightly-focusing regime.

The synthesis of the optical fields with a uniform polarization state is relatively straightforward, as it only involves shaping the phase and intensity of the beam. It can be achieved, for example, by using various types of holograms [10,11,12,13], spiral phase plates [14], digital micro-mirrors devices [15], or spatial light modulators (SLM) [16], which can provide almost any modulation of phase and amplitude. The above approaches rely mainly on expensive and advanced techniques involving a bulky setup comprising many different optical components. Moreover, synthesizing an optical field with complex polarization states requires superpositioning two or more field components with distinct polarization states [4].

One of the methods for synthesizing radially or azimuthally polarized light beams has already been described by Yamaguchi [17]. He used a nematic liquid crystal (NLC) confined in an NLC cell in a sandwich-like configuration. The NLC cell consisted of two glass plates specially treated with a rubbing technique. One of the inner surfaces of the NLC cell was rubbed uniformly in a particular direction, while the second one was rubbed circularly. As a result of the spatial-varying molecular twist pattern along with the cell thickness, the initially linearly polarized light incident from the side of a uniform alignment (the input glass plate) toward the side of a circular alignment (the output glass plate) converts into an axially symmetrical polarization state of the first order or topological charge, such as azimuthal and radial. Optical beams with higher-order polarization states, wherein the polarization vector rotates more than once within the angular coordinate range from 0 to 2π, can be generated in setups consisting of NLC devices with axial molecular alignment, as presented in [18,19]. To date, multiple realizations of such a method have been raised, including the usage of the NLC cells with circular-shaped electrodes [20], the photoalignment technique [21,22], cylindrical polymer cavities [23], or spontaneous formation of the periodic defect array initiated by standing pressure wave [24]. In recent years, such passive NLC polarization converters have been less attractive due to the increasing popularity of SLMs, which allow for more flexibility in complex beam shaping [25,26]. A spatial light modulator is a liquid crystal device that can operate as a programmable hologram. Due to their great flexibility in beam shaping, SLM can be applied for polarization and intensity shaping and phase conversion, especially for optical vortices generation. However, it is not easy to integrate SLM into compact optical systems, especially integrated ones, since realizing the conversion of all three mentioned light parameters requires more extensive and complex optical setups [27,28]. Additionally, operating an SLM is more complicated and requires more advanced knowledge than utilizing passive elements, such as phase structures or vortex retarders. That is why such devices are used mainly to generate more complex and non-trivial types of structured light.

To generate a vector vortex beam of a low TC (1 or 2), the so-called q-plates are commonly used. They are spatially variant, liquid crystal half-wave retarders characterized by the rotation of the optical axis around the center of the structure. Although similar elements can be formed by simply merging multiple typical half waveplates [29], the q-plates are used due to their manufacturing simplicity and the flexibility of NLC. Such optical structures can convert a linearly polarized Gaussian beam into an optical vortex of cylindrically symmetric polarization or rotate a circular polarization of the initial beam [30,31]. However, the input polarization state entirely defines the output state of a beam transmitted through the q-plate. Thus, for generating optical vortices of arbitrary polarization, it is simpler to combine the NLC polarization converter (NLC-PC) with polarization-independent methods of phase rotation, such as non-liquid crystal spiral phase plates [21,32,33].

Although vortex beams are associated with optical wavelengths mostly, it is also worth noting that they can be generated within a broader spectrum of E-M waves, further increasing their usability. One example is microwave beams carrying orbital angular momentum achieved using a reflect array antenna, which can find applications in short-range wireless and satellite communication [34,35].

This paper presents the concept of a compact programmable optical device designed to synthesize optical vortices with the desired polarization state. To verify our idea, we prepare a prototype that comprises a nanostructured vortex mask characterized by gradient refractive index distribution and a liquid crystal polarization converter. Light conversion occurs in a two-stage process: (i) first, using a sequence of three liquid crystal cells, a linearly polarized Gaussian beam is converted to the desired output polarization state, and then (ii) using a nanostructured vortex mask, it is converted into an optical vortex without any change in its polarization state. The nematic liquid crystal polarization converter comprises: (1) the NLC θ-plate for polarization conversion to an axially symmetric state, (2) an NLC phase retarder used for proper operation of the NLC θ-plate, and (3) a homogeneously oriented planar NLC cell that is a tunable phase plate. Combining these three elements in a single optical circuit enables the conversion of a linearly polarized input beam to one of eight different output polarization states: linear (parallel or perpendicular), elliptical (with two turns), azimuthal, radial, and vortical (with two turns). Switching between the different operation modes, i.e., the different beam polarization at the output is realized by a low-frequency external electric field (1 kHz). Due to its compact size and the possibility of electrical control, such a device can be integrated with existing optoelectronics components allowing for fast switching between the different SOP at the output.

## 2. Materials and Methods

The transformation of a Gaussian beam into an optical vortex is realized using a nanostructured gradient index vortex mask (nGRIN VM) made of two types of borosilicate glasses, labeled as NC32 and NC21, characterized by different refractive indices. The VM operation principle can be described by the effective medium theory (EMT) [33,36]. The scheme of the analyzed vortex structure is presented in Figure 1a. It contains 101 rods placed on the diagonal. The diameter of each rod diameter is equal to d = 0.208 μm, which fulfills the subwavelength structure condition at the designed wavelength λ = 1064 nm.

The internal structure of the nGRIN VM is determined by the distribution of low- and high-index nanorods inside the structure and is calculated using the simulated annealing (SA) method [37]. This method calculates the effective refractive index distribution for the lens using EMT, inspired by the basic Maxwell–Garnet equation, for a given rod distribution in the structure [38,39]. The glasses used have a refractive index depending on the incident wavelength. A vortex mask is designed to operate at a near-infrared wavelength of Nd:YAG laser at λ = 1064 nm. For the difference in refractive indices equal to ∆n ∝ 0.025 (n_NC32_ = 1.5401 and n_NC21_ = 1.5154, at λ = 1064 nm), a vortex structure length that provides a phase difference of 2π is about 43.1 μm. For the convenience of the experimental investigation, the vortex mask and a glass holder were glued together to a typical microscope glass slide. A microscopic image of the analyzed vortex mask is depicted in Figure 1b. The VM structure has a hexagonal aperture of about 21 ± 1 μm at the diagonal and is embedded into a glass holder with a diameter of 125 µm made of NC21 glass.

The VM mask was tested with a collimated, linearly polarized Gaussian beam with a wavelength of λ = 1064 nm and a radius in the focal point w_0_ ≅ 3 μm. A set of two polarizers and half-wave plates were used to adjust the input beam’s power and the azimuth of linear polarization. The beam was focused on the center of the mask, as schematically presented in Figure 1c. The transmitted light was collected by a 20× microscope objective (NA = 0.42) that formed an approximately non-diverging vortex beam with a diameter of about two millimeters. The visualization of a phase singularity was realized due to a vortex beam superposition with a reference in the Mach–Zehnder (M-Z) interferometer. The M-Z interferometer also contains a delay line to match the optical distance for the vortex and reference beams to maximize the contrast of interference fringes at the output. The CCD camera was used to visualize the output’s resultant optical field.

The proper operation of the VM was verified for two states of linear polarization—horizontal and vertical, as presented in Figure 1d,e. The left panels of Figure 1d,e show an intensity distribution of the generated vortex. Independently of the input polarization, they exhibit a typical doughnut-shaped pattern. The photos in the middle panels of Figure 1d,e result from a vortex interference with a reference beam. A characteristic fork-type pattern phase singularity with one additional interference fringe indicates the charge of the vortex of *m* = 1 for both polarizations of the input beam. The polarization distribution of the generated vortex beam was determined using the Stokes method. We found Stokes vectors for every pixel of a CCD camera for each output beam polarization. The calculations of the Stokes vector were based on six pictures of the light passed through different polarization filters: horizontal, vertical, diagonal, anti-diagonal, right-circular, and left-circular [40]. The polarization distribution of the vortex beam is presented in the right panel of Figure 1d,e in the form of lines representing an averaged area of 70 by 70 pixels [41].

The results indicate that the Gaussian beam is transformed into a vortex with an unchanged polarization state. It confirms that the VM affects only the phase of the beam. Based on the experimental findings, it was shown that the analyzed nanostructure is well-designed and operates correctly at a near-infrared wavelength.

## 3. Results

### 3.1. NLC Polarization Converter

The input polarization of the input beam was controlled by a liquid crystal polarization converter (LC-PC). By using three different NLC cells, presented schematically in Figure 2a, we can transform a linearly polarized beam into one of eight other typical SOPs: linear (parallel and perpendicular to the input), circular (right-handed, left-handed), or axially symmetric (azimuthal, radial, vortical left- and right-handed). Each cell has transparent indium–tin–oxide (ITO) electrodes that enable the out-of-plane reorientation of NLC molecules. The electrodes are covered with a specially prepared NLC alignment layer using a rubbing technique. The sketch of the molecular orientation and geometry of the electrodes is presented in Figure 2b–d. The first is a π phase retarder, the second acts as a steerable waveplate, and the last one, the θ-plate, transforms the uniform polarization into a cylindrically symmetrical SOP.

For clarity, we start the description from the middle cell presented in Figure 2c. It possesses a uniform planar orientation of molecules aligned at π/4 to the transmitted beam’s initial polarization direction along the *y*-axis. The ITO layers fully cover both glass plates of a cell. By changing the amplitude of the U_2_ voltage applied to the electrodes, we can change the optical properties of the nematic liquid crystal structure to act as a: 0 λ—waveplate (no change in polarization), λ/2—waveplate (perpendicular polarization at output), λ/4—waveplate (linear to circular polarization), and 3/4 λ—waveplate (linear to circular polarization with the opposite handedness to the λ/2—waveplate mode). The proper U_2_ voltages were defined experimentally by analyzing the light transmission of the λ = 1064 nm infrared beam in a configuration where the NLC waveplate was put between crossed polarizers. The following relative retardance between two orthogonal polarization axes (*x* and *y*) was recorded: no retardance at U_2_ = 1.3 V, π/2 at U_2_ = 2.2 V, π/4 at U_2_ = 3.3 V, and 3/4 π at U_2_ = 1.6 V.

Conversion of the linear polarization into an axially symmetrical SOP requires a more complex structure called θ-cell, schematically presented in Figure 2d, with an additional π-phase retarder, as sketched in Figure 2b. The θ-cell is combined with two glass plates possessing different anchoring conditions for NLC molecules. At the input, the molecular director exhibits a uniform orientation along the *y*-axis (as indicated by solid black lines in Figure 2d). At the output, the molecular alignment is circularly symmetrical (dashed lines, Figure 2d). Therefore, a twisted nematic pattern with a spatially variant local twist within the range of 〈−π/2, π/2〉 is formed as a result of the minimization of the elastic twist energy. The cell’s thickness (12 μm) ensures fulfilling the Maugin condition [42] for the used nematic liquid crystal (6CHBT; birefringence ∆n = 0.15 [43]), and the polarization direction of the transmitted beam follows the molecular twist. As a result, a transmitted beam characterized by the linear polarization at the input is transformed into one of the axially symmetrical SOPs: azimuthal, radial, or vortical, depending on the azimuthal direction of the beam polarization vector. By applying a high enough U_3_ voltage, the molecules tend to reorient toward the *z*-axis (to homeotropic configuration). Consequently, the structure is switched off, and no polarization change for the transmitted beam is realized. Because the local twist angle in the NLC cell cannot be larger than ±π/2, a uniform-to-circular molecular twist between opposite cell substrates results in the appearance of two halves with opposite twist directions. The border between both areas is visible as a line defect when an NLC cell is observed between crossed polarizers. It is caused by an additional π phase delay of propagated light by adjacent areas [19]. This affects the polarization conversion leading to a degeneration of the output polarization state. The solution to this problem is to introduce a proper phase delay of the input beam. It is realized by an additional NLC cell presented in Figure 2b that acts as a tunable phase retarder. This cell features a homogenous arrangement of NLC molecules along the *y*-axis, the same as the input plate of the θ-cell; however, the electrodes are deposited in only half of the cell. Applying the voltage creates a uniform electric field; thus, only part of the molecules are affected by it, changing their orientation from the *xy*-plane towards the *z*-axis. Because of the voltage-induced reorientation of the molecules, the transmitted light “sees” only the change in the effective refractive index-inducing phase without the polarization conversion. To acquire the desired π-phase delay in the designed cell, we need to apply the voltage of U_1_ = 2.3 V, which was determined experimentally in the interferometric configuration as a value for which two halves of the beam are in the counter phase.

By connecting the three structures described above into one NLC-PC device, as shown in Figure 2a, it is possible to convert the linear polarization of the Gaussian input beam (let us assume horizontal direction for clarity along the *y*-axis) into one of eight different SOPs by applying proper U_1_, U_2_, and U_3_ voltages to each cell. The exact voltages are summarized in Table 1. Switching time is typical for liquid crystal cell devices and does not exceed a few tens of milliseconds. The molecular reorientation time directly depends on the amplitude of the driving voltage. The reorientation process occurs significantly faster for a driving voltage with an amplitude higher than the threshold value, and NLC structures can be switched within a single millisecond [44].

The final NLC-PC device was verified experimentally, as shown in Figure 3a,b. By changing the U_1_, U_2_, and U_3_ voltages, we can switch the operating modes of each cell (between “on” and “off” state for the phase retarder, the θ-cell, and the waveplate), the input linear polarized Gaussian beam can, therefore, be converted into one out of eight different SOPs: linear (parallel and perpendicular to the input), circular (both handedness), and cylindrical symmetrical (azimuthal, radial and vortical both handedness). In the latter case, shown in Figure 3b, a central point of low intensity is seen, giving the beam a doughnut-shaped profile of a vector beam; however, it appears due to the polarization singularity only, and the phase remains unchanged. Due to averaging Stokes’ vector’s values over a significant pixel, the resulting polarization distribution images, especially for vector beams, are slightly disturbed; however, their overall shapes are distinguished, and polarization handedness, for cases of the circular polarization at the output, are uniformly distributed across the beam, proving the proper operation. In the latter, elliptical polarization seen on the beam edges results from the measurement method’s vulnerability to the spatial shift of each polarization component image.

### 3.2. Polarization and Phase Conversion

Presented NLC-PC and VM were merged into one device, as schematically shown in Figure 4a, to be used as a vector vortex beam generator with a programmable state of polarization. The proposed device can change a linear-polarized Gaussian beam into an optical vortex with charge *m* = 1 and one of eight different SOPs. The polarization-independent feature of the nanostructure VM enables easy integration of both modules.

The proposed device was tested in a typical Mach–Zehnder interferometric setup similar to the one presented in Figure 1c. Figure 4b–e illustrates the generation of the first-order vortex beams from a linear polarized Gaussian beam of 1064 nm wavelength, characterized by one of the seven different SOPs at the output. As shown in Figure 4b,c, the recorded output beam exhibits a clear doughnut-shaped intensity distribution and characteristic fork-like intensity fringes when interfered with the Gaussian input beam, indicating proper transformation into a high-quality optical vortex beam. The output SOP also exhibits good quality. Linear and circular polarization states are uniform over the entire cross-section, as shown in Figure 4d, first two panels from the left; for cylindrical symmetrical SOP, the appropriate symmetry is also visible, as shown in Figure 4d, first two panels from the right and the bottom ones.

For clarity, Figure 4 presents the results obtained only for a specific linear polarization) at the input (horizontal, along the *y*-axis) for which our proposed converter has the broadest range of possible output SOP. To obtain an arbitrary linear polarization state at the output (any between horizontal and vertical one), we would need to include an additional planarly aligned NLC cell or insert a half-wave or quarter-wave plate, respectively. In such an extended configuration, it would also be possible to get elliptical polarization at the output, with the ellipticity depending on the applied voltage.

Nevertheless, using the device as presented in Figure 4a, a limited conversion from vertical linear (along the *x*-axis) and circular SOP is achievable. Experimental results are depicted in Figure 5. The pictures in Figure 5a–d correspond to the linear vertical polarization of the input beam, and the ones in Figure 5e–h correspond to the initial left-handed circular SOP. The results indicate that a linear vortex beam (horizontal, vertical, and ±45°) can be obtained at the output for linear vertical or circular input SOP. They are further increasing the spectrum of possible SOP outputs. However, when using circularly polarized input, an additional NLC cell would have to be inserted at the proposed device input to achieve a vector vortex beam to convert the circular polarization into a linear one.

## 4. Conclusions

In conclusion, we have presented and tested a structure prototype for generating optical vortices with a programmable polarization state at the output that operates with a linearly polarized Gaussian beam. The proposed optical device enables intensity and polarization beam shaping. The operation principle utilizes the NLC polarization converter combined with an especially designed nanostructured gradient index vortex mask working as a phase shaper. With the proposed device, an input Gaussian beam with linear polarization can be effectively transformed into a compact optical vector vortex beam with one of eight SOPs: linear vertical and horizontal, circular left and right-handed, azimuthal, radial, and vortical. Furthermore, it is also possible to use an input beam with circular polarization and obtain an output vector vortex with two additional uniform linear SOPs. In both cases, tuning between different output polarizations is accomplished by applying an external electric field @1 kHz. The current state of technology allows us to assume that realizing our programmable vortex beam converter proof-of-concept would be feasible with all-fiber technology and 3D direct laser printing [45,46] and optical fibers combined with embedded electrodes [47].

Moreover, the development of the device with the additional planarly aligned NLC cell into the proposed device will enable it to also operate with arbitrary output polarization (linear with any azimuthal angle, circular left- and right-handed, elliptical with any degree of ellipticity, radial, azimuthal, and vortical). That means that the proposed device can be widely used, and the ease of output polarization tuning is a clear advantage over other solutions. Its small size and electrical control capability allowing easy integration within existing optical setups, make such a device an excellent option to fill the gap between popular methods of complex beam shaping. It does not require a significant extension of the system nor professional knowledge to operate, which are the major issues in using SLM; also it is a cheaper solution. It also allows for fast output modulation without additional elements, unlike the popular q-plates. Thus, in our opinion, the presented device can play a compact and low-cost beam (intensity and polarization) shaper in areas such as laser processing and machining, where the simplicity of operation and reliability are more important than the ability to generate a wide range of structured beams.

## Figures and Tables

**Figure 1 materials-15-07794-f001:**
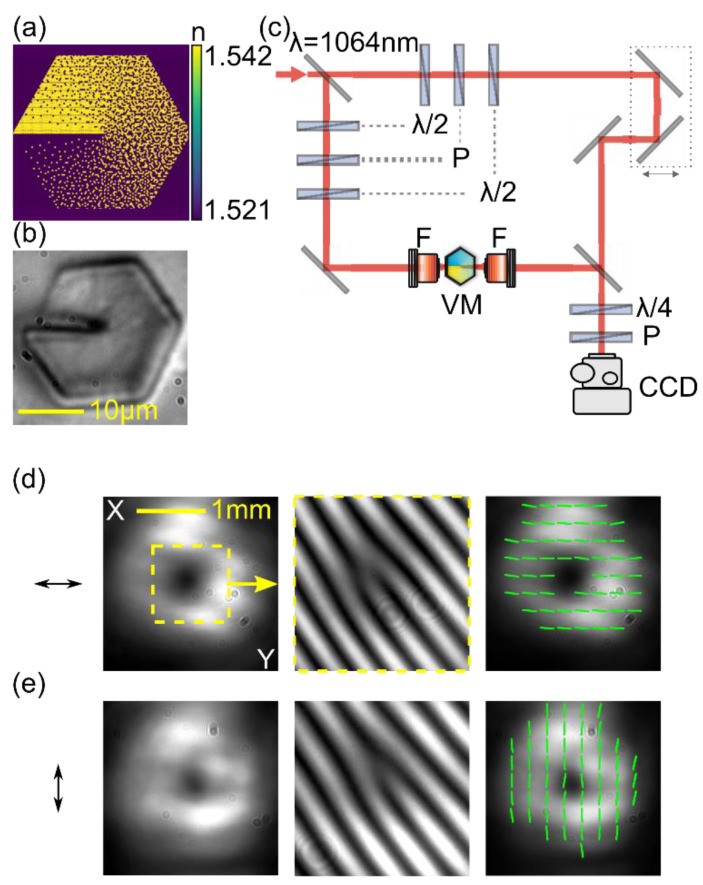
(**a**) Schematic representation of the vortex nanostructure with 101 rods on the diagonal. Yellow (violet) dots represent high (low) refractive index rods; (**b**) Photo of the examined structure; (**c**) Sketch of the experimental setup: VM—vortex mask, F—lens, CCD—digital camera; (**d**) Experimental results of Gaussian beam transformation through the examined structure. The left, middle, and right panels represent the output intensity distribution, phase singularity within the marked area, and polarization distribution, respectively. Obtained for the wavelength of λ = 1064 nm and linear polarization (horizontal direction—schematically marked by the black arrow); (**e**) Experimental results of Gaussian beam transformation for a linearly polarized input beam, vertical direction—orthogonal to the input beam in (**d**).

**Figure 2 materials-15-07794-f002:**
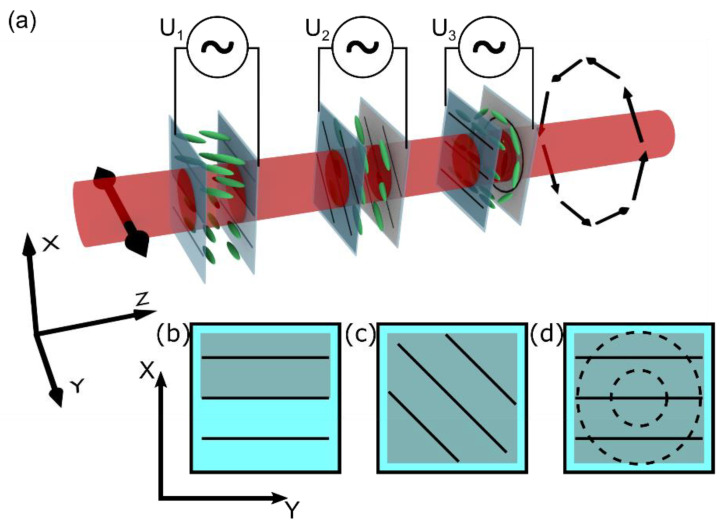
(**a**) Schematic representation of the NLC polarization converter composed of three independent NLC cells, controlled by the external voltage; (**b**) π-phase retarder; (**c**) waveplate; and (**d**) θ-plate. Schematic representation of each cell presented in (**b**–**d**): solid (dotted) black lines indicate the alignment of molecules at the input (output) side, grey areas—ITO layer.

**Figure 3 materials-15-07794-f003:**
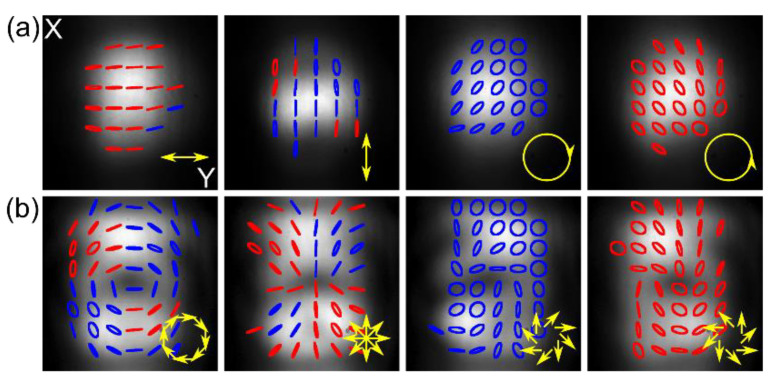
The experimental results of converting the *y*-polarized Gaussian beam using NLC-PC into (**a**) uniform and (**b**) cylindrically symmetric SOP. Blue and red represent right- and left-handedness, respectively, and yellow arrows show the desired output SOP achieved by applying proper U_1_, U_2_, and U_3_ voltages, as summarized in Table 1.

**Figure 4 materials-15-07794-f004:**
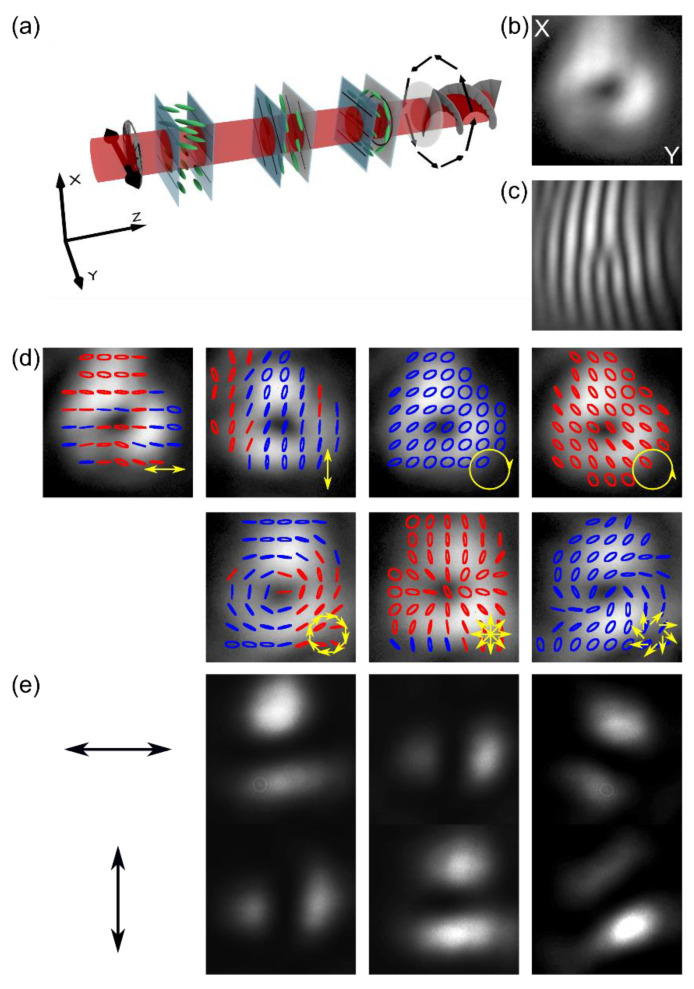
(**a**) Schematic representation of Gaussian-vortex polarization converter consisting of NLC-PC and VM; The result of linearly polarized Gaussian beam transformation into an optical vortex: (**b**) doughnut-shaped intensity profile of a linear (horizontal) output polarization, (**c**) fork-like interference fringes; (**d**) the result of the polarization conversion of the vortex beam into seven different SOPs (from the left-top panel): linear horizontal, linear vertical, circular right- and left-handed, azimuthal, radial, and vortical; (**e**) horizontal (top panels) and vertical (bottom panels) linear polarization components (schematically marked by the black arrows) of azimuthal, radial, and vortical SOPs.

**Figure 5 materials-15-07794-f005:**
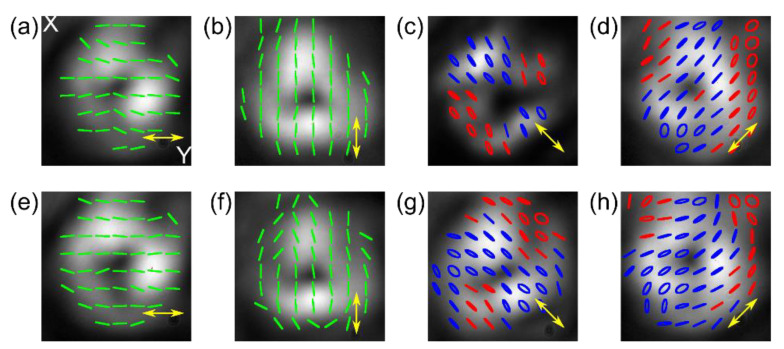
The intensity and polarization distribution of generated vector vortex beams that result from the conversion of (**a**–**d**) linear (vertical) polarized Gaussian input beam; (**e**–**h**) left-handed circular polarized input Gaussian beam for U_1_ = 0 V, U_3_ = 5 V driving voltages, and for the following U_2_ voltages: (**a**) U_2_ = 2.2 V; (**b**) U_2_ = 1.3 V; (**c**) U_2_ = 1.6 V; (**d**) U_2_ = 3.3 V; (**e**) U_2_ = 3.3 V; (**f**) U_2_ = 1.6 V; (**g**) U_2_ = 2.2 V; (**h**) U_2_ = 1.3 V.

**Table 1 materials-15-07794-t001:** The driving voltages for the NLC-PC device that results in the desired output SOP for the transmitted beam are presented for the Gaussian linear polarized beam of wavelength λ = 1064 nm.

Cell	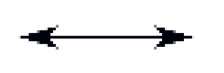	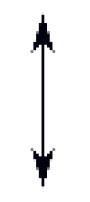	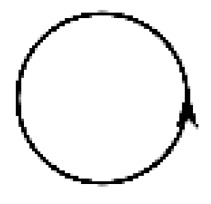	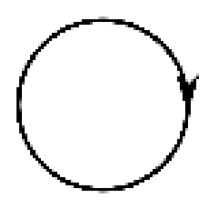	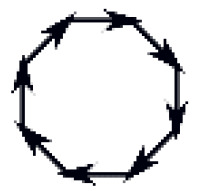	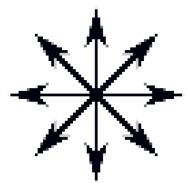	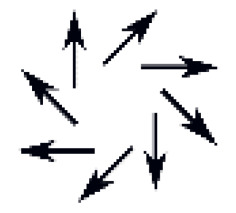	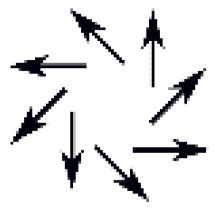
Phase retarder—U_1_ [V]	0	0	0	0	2.3	2.3	2.3	2.3
Waveplate—U_2_ [V]	1.3	2.2	1.6	3.3	1.3	2.2	1.6	3.3
θ-plate—U_3_ [V]	5.0	5.0	5.0	5.0	0	0	0	0

## Data Availability

Not applicable.

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
