# Peer review of "A Gaussian to Vector Vortex Beam Generator with a Programmable State of Polarization"

_materials, 2022, doi:10.3390/ma15217794_

Round 1
Reviewer 1 Report
See attached file

Author Response
Dear Reviewer,
Thank you very much for all your valuable comments; however, we do not entirely agree with the suggestion of lack of novelty. Methods described in the mentioned article are based on one or more mechanically rotating parts, which are not as reliable as electrically steered nematic liquid crystal structures. Our polarization converter demonstrator can be further minimalized even to the size of tens/hundreds of microns, thanks to the well-developed technology of liquid crystal materials and, for example, nano 3D printing technology. A final device would allow for significantly easier integration with integrated optical systems, supporting SOP switching time in milliseconds. The additional and essential thing is the possibility of electrical tuning presented polarization converter to desired wavelengths.
Although our method is based on already known principles, it presents some advantages over similar methods for light polarization conversion. We hope that provided explanation will be sufficiently comprehensive to reconsider the review.
Best regards,
Michal Kwasny
Reviewer 2 Report
The authors proposed a vortex generator with tunable polarization. The generator consists of three nematic liquid crystal cells (a π-phase retarder, a waveplate, and a θ-plate) and a vortex mask. By changing the applied voltage of cells, the linearly polarized input beam can be converted into one of eight different output polarization states. I suggest the manuscript should have major revisions before publication. The comments are as follows:
1. The states of polarization of output beams need to be detected experimentally with polarizer, especially for radially and azimuthally polarized beams. Corresponding experimental results should be added.
2. What do the blue and red lines represent respectively in Fig. 3? The calculation method for determining polarization distribution by using stokes parameters needs to be given in detail.
3. Why is circular and elliptical polarization mixed in the polarization distributions of Fig. 3? Can this be avoided?
4. As a comparison, the results of right-handed polarization should be added in Fig. 4.
5. Explain in detail what the probes on the left and right sides of the VM in Fig. 1(c) are and their functions.
6. What is ‘nGRIN’? There is no explanation in the manuscript. In addition, the manuscript needs to be proofread. For example, when the first time ‘spatial light modulator’ appeared, it was not abbreviated.
Author Response
Dear Reviewer,
Thank you very much for all your valuable comments. In the revised version of the article, we addressed the issues mentioned above. They are concerned, in particular, with the following statements:
- The states of polarization of output beams need to be detected experimentally with a polarizer, especially for radially and azimuthally polarized beams. Corresponding experimental results should be added.
Our polarimetry method is based on calculations over output intensity profiles in a setup with a quarter-wave plate and polarizer, which is a more general approach. Nevertheless, for clarity, we added to fig.4 images of two opposite linear polarization components of cylindrical symmetric polarized beams.
- What do the blue and red lines represent, respectively in Fig. 3? The calculation method for determining polarization distribution by using stokes parameters needs to be given in detail.
We have overlooked this information. The description was added in the caption of Figure 3.
- Why is circular and elliptical polarization mixed in the polarization distributions of Fig. 3? Can this be avoided?
We expanded the commentary regarding the disturbing polarization profiles due to the inaccuracy of the measurement method.
- Explain in detail what the probes on the left and right sides of the VM in Fig. 1(c) are and their functions.
We added the description in the caption. The "probes" are converging lenses.
- What is 'nGRIN'? There is no explanation in the manuscript. In addition, the manuscript needs to be proofread. For example, when the first time' spatial light modulator' appeared, it was not abbreviated.
The explanation of nGRIN was added (nanostructured gradient index vortex mask).
We have also proofread the manuscript more carefully.
I hope that the provided explanations will be sufficiently comprehensive and enable our work's publication.
Best regards,
Michal Kwasny
Reviewer 3 Report
See the attached file.

Author Response
Dear Reviewer,
Thank you very much for your positive review as well as valuable comments. In the revised version of the article, we addressed the issues mentioned above. They are concerned, in particular, with the following statements:
- But the effective medium is not even mentioned in reference 35! So, the authors must come up with references containing the application of the theory of effective medium to the VM structure, or as the case may be, the dielectric mixing equations.
This omission makes it very difficult to understand the physical reasons for the operation of the VM, and the authors should provide sufficient information concerning the underlying theory of the VM!
We have updated the references to ones describing the application of EMT to VM structures.
- Scalar Several methods like the Bruggeman, Raleigh, Maxwell-Garnett are used if it comes to “effective medium theory”. Which one is used??
See e.g., Effective Medium Theory Principles and Applications, Second Edition, TUCK C. CHOY, and: Ref. 36 Electromagnetic Mixing Formulas and Applications, by Ari Sihvola. Published by The Institution of Engineering and Technology, London, United Kingdom
We have added information about using the Maxwell-Garnett method along with the references containing further descriptions. We also include more in-depth commentary at the end of our response.
I hope that the provided explanations will be sufficiently comprehensive and enable our work's publication.
In the attachment there are some additional informations about the effective medium theory (EMT).
Best regards,
Michal Kwasny

Reviewer 4 Report
In the submitted manuscript the Authors propose an interesting solution to convert a gaussian beam into an optical vortex with different polarization states. The effectiveness of the proposed solution is proved with experimental results and the manuscript is generally well written. I recommend the publication of the manuscript after the following minor changes.
1) The manuscript is in general well written but some typos should be fixed. See, for instance, "fulfils"
2) Scalar and vector vortex beams have been exploited also at microwave frequencies. I suggest slightly extending the state-of-the-art review also to microwave frequencies to broaden the possible interested readership. See, just for instance:
- M. Veysi, C. Guclu, F. Capolino and Y. Rahmat-Samii, "Revisiting orbital angular momentum beams: Fundamentals reflectarray generation and novel antenna applications", IEEE Antennas Propag. Mag., vol. 60, no. 2, pp. 68-81, Apr. 2018.
- H. Huang and S. Li, "High-Efficiency Planar Reflectarray With Small-Size for OAM Generation at Microwave Range", IEEE Antennas Wirel. Propag. Lett, vol. 18, no. 3, pp. 432-436, March 2019.
- M. Barbuto, A. Alù, F. Bilotti and A. Toscano, "Dual-Circularly Polarized Topological Patch Antenna With Pattern Diversity," in IEEE Access, vol. 9, pp. 48769-48776, 2021
3) A brief discussion about the commutation time between the different states should be added. Moreover, a clearer comparison about the thickness of the proposed approach and the one of other solution would be useful.

Author Response
Dear Reviewer,
Thank you very much for all your valuable comments. In the revised version of the article, we addressed the issues mentioned above. They are concerned, in particular, with the following statements:
- The manuscript is in general well written but some typos should be fixed. See, for instance, "fulfils"
The manuscript was again carefully proofread in search of typos.
- Scalar and vector vortex beams have been exploited also at microwave frequencies. I suggest slightly extending the state-of-the-art review also to microwave frequencies to broaden the possible interested readership.
A brief paragraph concerning the generation of OAM-carrying microwaves was added to the introduction.
- A brief discussion about the commutation time between the different states should be added. Moreover, a clearer comparison about the thickness of the proposed approach and the one of other solution would be useful.
Information about it was added alongside the reference to another article containing a more detailed discussion about commutation time in this type of NLC cells.
I hope that the provided explanations will be sufficiently comprehensive and enable our work's publication.
Best regards,
Michal Kwasny
Round 2
Reviewer 1 Report
Thank you very much for clarifying to me the advantages of your proposal over other systems for changing the polarization state of light.
Author Response
Dear Reviewer,
Thank you again for the detailed review of our manuscript and for appreciating the submitted explanations. In addition, minor language errors have also been corrected in the latest version of the article draft.
Regards,
Michal Kwasny
Reviewer 2 Report
The authors used semi-analytical method, i.e. using scattering to process the experimental data to obtain the vector profile, why not use the method taken in the following work? It is a direct, simple and accurate method to measure radial and azimuthal polarization. [Figure 3 in Applied Physics Letters, 2000, 77(21): 3322-3324; Figure 4 in Optics Express, 2010, 18(10): 10786-10795; and Figure 5 in Photonics Research, 2016, 4(5): B35-B39.].
Author Response
Dear Reviewer,
We agree that linear polarization components of generated optical fields would also be presented with the use of a more straightforward approach. However, the experimental results concern many vector vortex beams with different polarization states. Presenting them with the mentioned method would result in a considerable increase of elements in most figures (at least five times more in Fig. 3-5). In our opinion, it would make the manuscript much harder to read, thus decreasing the results' clarity. Considering that, we decided to present polarization maps, which although having their flaws, are much more readable for comparing different vortex beams.
Regards,
Michal Kwasny